# Response Surface Methodology for the Optimization of Ultrasound-Assisted Extraction of Tetrodotoxin from the Liver of *Takifugu pseudommus*

**DOI:** 10.3390/toxins10120529

**Published:** 2018-12-10

**Authors:** Xiaojun Zhang, Chengcheng Han, Si Chen, Le Li, Jingjing Zong, Junjie Zeng, Guangming Mei

**Affiliations:** 1Laboratory of Aquatic Product Processing and Quality Safety, Marine Fisheries Research Institute of Zhejiang, Zhoushan 316100, China; sichen_ns@zjou.edu.cn (S.C.); zengjunjie0000@zjou.edu.cn (J.Z.); meigm123@zjou.edu.cn (G.M.); 2School of Food and Pharmacy, Zhejiang Ocean University, Zhoushan 316022, China; hanchengcheng417@163.com (C.H.); zongjingjing1210@163.com (J.Z.); 3Quality and Standard Research Center, Chinese Academy of Fishery Sciences, Beijing 100141, China

**Keywords:** tetrodotoxin, response surface methodology, pufferfish

## Abstract

Tetrodotoxin (TTX) is a marine biotoxin that has high scientific value. However, the lack of efficient TTX extraction and preparation methods has led to a scarcity of TTX samples for clinical application. In this study, TTX from the liver of *Takifugu pseudommus* was ultrasound-assisted extracted with acidified organic solvents. The extraction process was analyzed and optimized by single factor method and response surface methodology (RSM). The optimal extraction conditions predicted by a response surface model were as follows: liquid:material ratio, 2.8:1; extraction temperature, 60 °C; extraction time, 23.3 min. Under these conditions, the extraction of TTX had a yield of 89.65%, and the results were further verified by experimental extraction, and analyzed by ultra performance liquid chromatography–tandem mass spectrometry (UPLC–MS/MS). It was found that the extracts of *T. pseudommus* liver contained TTX and its four analogues at certain proportions (TTX: 10.4%; 5,6,11-trideoxyTTX: 83.3%; 5,11-dideoxyTTX:2.4%; 4,9-anhydro TTX:2.6%; 5-deoxyTTX:1.3%). This study demonstrates a stable and efficient extraction process of TTX from pufferfish liver, which can be helpful for further research and analysis, as well as the utilization of TTX from pufferfish.

## 1. Introduction

Tetrodotoxin (TTX) is a low-molecular-weight marine biotoxin that is one of the most powerful natural non-protein neurotoxins ever known [1]. As shown in Figure 1, the structure of TTX consists of a guanidinium moiety connecting to a highly oxygenated carbon skeleton that is composed of a 4-dioxaadamantane containing five hydroxyl groups [2]. TTX was first discovered in pufferfish in 1964 [3]. Since then, it was also discovered in gastropods [4,5], newts [6,7], crabs [8], sea slugs [9], star fishes [10], and other marine organisms. Triggered by blocking site 1 of the voltage-gated sodium channel, the human intoxication syndrome of TTX is characterized by symptoms of respiratory paralysis and in the most severe cases, death. Because of its unique role in sodium ion channels, TTX can be used clinically for local anesthesia [11] and detoxification [12], and as a labor pain relief [13]. There are no obvious side effects of TTX [14], therefore, it may have a great potential in medical application. Therefore, a number of studies have largely been devoted to extracting TTX from marine organisms [7] or to synthesizing TTX [15] in sufficient amounts for medical and food safety research.

Currently, TTX extraction methods mainly include microbial fermentation [2] and biological tissue extraction [13]. The microbial fermentation method can only yield nanogram amounts of TTX, and it is difficult to obtain a larger amount, due to the low productivity of the method. On the other hand, the bio-extraction is the most extensively studied method for preparing TTX with larger quantities. It was traditionally and mainly used to extract TTX from pufferfish by using aqueous acetic acid solutions [16,17]. However, since water is not easily evaporated or concentrated, this can subsequently bring a number of problems to the purification process of TTX, and can thereby affect the extraction efficiency. It is necessary to improve and optimize the critical extraction process to promote the TTX production. According to our trail experiment, methanol containing 1% acetic acid served as an efficient substitute for acidified aqueous solution, yet systematic optimization has not been reported.

In recent years, some new extraction techniques, such as ultrasound-assisted extraction (UAE), microwave-assisted extraction, and accelerated solvent extraction [4] have been gradually applied to the isolation and extraction of marine biological active substances. Among these methods, the UAE method extracts the active ingredient by increasing the moving speed and the penetration of the medium molecules, which is achieved due to its unique mechanical, cavitational, and thermal effects. This method is also fast, simple, and efficient [18].

As an effective experimental design method, response surface methodology (RSM) can be carried out with multiple factors and levels. It can be used to study the interaction between factors, and to obtain optimal processes and results, and thus is more advantageous than other design methods, such as orthogonal experiments [19]. Currently, RSMs have been widely used for the industrial optimization and manufacturing of various functional products in the field of biotechnological and biochemical applications [20]. The application of RSM in the optimization of the extraction and the preparation of marine biotoxins also has good prospects.

In this study, TTX from pufferfish liver was extracted by the UAE method by using acetic-acidified methanol as the organic solvent, and the process was optimized by RSM to establish a high-efficiency extraction method. The study presents a highly efficient and stable TTX extraction method, which can not only be beneficial to the scientific community, but also may encourage the utilization of TTX in the medical and food safety fields.

## 2. Results and Discussion

### 2.1. Analysis of Single Factor Optimization

For a strong polar substance TTX, a 0.1% acetic acid solution was extensively used to extract the analyte from biological samples [21]. However, a high solubility of proteins and carbohydrates in water usually led to a cloudy extracting solution, which was hard to concentrate into a smaller volume, subsequently posing a challenge for further TTX purification. A high percentage of organic reagent in the extraction system generally permits effective protein precipitation and carbohydrate removal. In this study, methanol was initially used as the substitution solution for water, and the effect of acetic acid concentration in methanol (%) on TTX extraction yield was evaluated. As shown in Figure 2, TTX recovery reached its maximum value of 96% when methanol contained 1% acetic acid. Therefore methanol containing 1% acetic acid was chosen as the ideal extraction solution in later experiments. The extraction of TTX can also be affected by a variety of important factors, including material:liquid ratio, extraction time, extraction temperature, and the number of extractions. Thereby, a systematic study needs to be investigated, to find out the most significant factor for TTX yield. The range of factors in turn improve the accuracy of the experiment. In this study, three single factors impacting TTX extraction yield, including material:liquid ratio (1:1 to 1:5), extraction time (10 to 40 min), and temperature (20 to 70 °C), were studied, and the results are shown in Figure 3. The experimental results showed that the extraction yield remained nearly constant at a material:liquid ratio of 1:3, and it was not significantly improved with the increase of the material:liquid ratio. Thus, a material:liquid ratio of 1:3 was selected for subsequent RSM analysis (Figure 3A). Furthermore, the TTX extraction yield reached the highest value at an extraction time from 20 to 25 min. It is possible the synergistic effect of various factors may improve the efficiency and in turn, reduce the extraction time. Thus, an extraction time of 15–25 min was selected for subsequent optimization (Figure 3B). As shown in Figure 3C, the extraction yield was stable at temperatures above 50 °C, indicating that TTX has excellent thermal stability. Considering both cost and time, an extraction temperature of 50 °C was chosen for subsequent optimization. For the effect of number of extraction, although the extraction yield gradually improved with increasing material:liquid ratio (Figure 3D), it did not significantly change after two cycles of extraction. Since an increased number of extraction could result in a larger volume of extraction liquid, consequently causing difficulty in sample concentration and purification, TTX was extracted twice with acidified methanol for subsequent experiments.

### 2.2. Fitting the Response Surface Models

Based on the results of the single factor experiment, RSM was used to further optimize the extraction of TTX from liver. Parameters including the material:liquid ratio, the extraction temperature, and the extraction time were set as the independent variables, and the TTX extraction yield was set as the response value. The three-factors–three-levels experiment was designed and carried out according to the principle of the Box–Behnken experimental design [22], and the results by means of TTX extraction yield are shown in Table 1.

The extraction yield of TTX, which was set as the response value, was used to perform multiple regression using Design expert 8.0.6 statistical software (Table 1). The obtained relationship between the influencing factors and the response value is as follows: R (the extraction yield of TTX (%) = 80.51 + 0.37×A + 0.57×B + 0.55×C + 0.57×A×B + 0.32×A×C + 0.47×B×C − 0.30×A^2^ − 0.90B^2^ − 0.96×C^2^.

The relation between each independent variable and response value described by the regression equation is shown in Table 2. The accuracy of the model was determined by the analysis of variance and correlation coefficient. According to the equation, the relationship between the dependent variable and all independent variables was linear. The *F* value was 6.68 with *p* < 0.05, suggesting that the method is accurate and reliable. The lack-of-fit item *p* value was 0.2698 with *p* > 0.05, implying that unknown factors have less influence on the experiment. A further test of the equation showed that the *F* value of the equation was 1.91, indicating that the lack-of-fit item was not significant relative to the pure error, and the error of the experiment was small. These analyses suggest that the model fitted well with the experimental results, and it thus can be used to accurately describe and predict the experimental results. As shown in Table 2, the effects of B, C, B^2^, and C^2^ on the results were significant, indicating that the relationship between each of these factors and the response value is not a simple linear relationship. The comparison of the *F* values (*F*_A_ = 3.75, *F*_B_ = 9.1, *F*_C_ = 8.55) implied that the material:liquid ratio was the most influential factor on the extraction yield of TTX, while the second factor was the extraction time.

### 2.3. Response Surface Analysis

The response surface curve is a three-dimensional spatial surface map that is composed of response values in relation to each experimental factor A, B, and C. The optimal parameters and the interaction between the parameters in the response surface curve can be seen more intuitively and visually [22]. The response surface curve, which was constructed based on the regression equation, is shown in Figure 4. Since the *p* values of AB, AC, and BC in the regression equation were higher than 0.05, the interactions between the liquid:material ratio, the extraction time, and the extraction temperature were not significant; and this can be intuitively observed in Figure 4A–C. The contour map shows that the condition at the center of the circle is the extreme value. By analyzing the steepness of three sets of curves, it can be concluded that the liquid:material ratio (B) was the most influential factor on the TTX extraction yield, followed by the extraction time (C) and the extraction temperature (A). Compared with the single factor and orthogonal experimental methods previously used in the extraction of TTX [23], the response surface analysis used in this study more intuitively and accurately showed the interaction between factors.

### 2.4. Optimization of UAE Process and Validation of the Model

According to the analysis and the prediction by the software, the optimal conditions, indicating the optimal values of each factor, for TTX extraction were obtained. The optimal conditions were as follows: liquid:material ratio, 2.81 g/mL; extraction temperature, 60 °C; extraction time, 23.25 min. Under these conditions, the TTX extraction yield could reach 90.91%. However, when the convenience of the actual extraction and operability of the experiment were taken into account, the parameter values were corrected, resulting in new optimal extraction conditions as follows: liquid:material ratio, 2.8; extraction temperature, 60 °C; extraction time, 23.3 min. To further confirm the accuracy and reliability of the optimal conditions, the actual TTX extraction experiment was carried out three times using the above optimal conditions. The results showed that the actual average TTX extraction yield was 89.65 ± 1.82%, which deviated from the model—predicted by only about 1.38%. The highly localized temperature and pressure provided by microwave-assisted extraction can cause selective migration of target compounds from the material to the surroundings solution at a more rapid rate and a higher extraction yield [24]. Compared with the traditional water-bath extraction method that utilizes an acidified aqueous solution [21], the new developed method had a significantly shortened extracting time, from 70 min to 23 min, as well as an improved TTX yield by 22%. Data indicates that the model can accurately predict the experimental optimal conditions, and thus it has high practical applicability and operability.

### 2.5. Identification of TTX

Recent studies have proven an abundant coexistence of various analogues of TTX [25,26]. Traditional method developed toxin extraction by targeting TTX as the main product [21], while lacking information on the identification and confirmation of the possible co-extraction of TTX analogues. In this study, a mathematical method describing the TTX extraction process was developed and used to predict the extraction of TTX from the liver of *T. pseudommus*. The extraction was performed in conjunction with the ultra performance liquid chromatography–tandem mass spectrometry (UPLC–MS/MS) analysis [27], which was used to further analyze and identify TTX analogues. The results indicated that there were various types of TTX in the liver of *T. pseudommus*, as confirmed by mass spectrometry (Figure 4). In addition to main TTX (Figure 5A), it also contained certain amounts of 5-deoxyTTX (Figure 5B), 4,9-anhydro TTX (Figure 5C), 5,11-dideoxyTTX (Figure 5D), and 5,6,11-trideoxyTTX (Figure 5E). These compounds co-existed with TTX in the liver of *T. pseudommus* at the following proportions: 10.4% (TTX); 83.3% (5,6,11-trideoxyTTX); 2.4% (5,11-dideoxyTTX); 2.6% (4,9-anhydro TTX); 1.3% (5-deoxyTTX). Since the traditional method was tedious and time-consuming, MS confirmation was only carried out for the pilot scale of acidified aqueous extract during the initial phase of method comparison, as mentioned above. TTX and another four analogues were also found in the aqueous extract; however, the extraction yields for these five compounds experienced a 22–43% reduction. Compared with TTX found from other organisms (gastropods [4], newts [7], pufferfish (*T. oblongus*) [1]), the types and contents of TTX found in this study are unique to the *T. pseudommus* liver. Study have shown that TTX analogues are distributed between either the tissues of the same organism, or that different specimens of the same species was unequal [27], Meanwhile, 5,6,11-trideoxyTTX was found to be the major analogue in all tissues of pufferfish and gastropods [1,4]. Considering the fact that the liver material was a combined sample consisted of different batches of pufferfish, our results were comparable to previous reports. Moreover, the mass spectra corresponding to TTX analogues found in this study can provide basic information for the further separation and preparation of TTX and analogues.

## 3. Conclusions

In this study, RSM was successfully applied to optimize the UAE process of TTX from the liver of *T. pseudommus*. Factors including the liquid:material ratio, extraction time, and extraction temperature were optimized by single factor and RSM optimizations, and TTX extracted under optimized conditions was analyzed by UPLC-MS/MS. The analysis showed that the extract contained TTX and four analogues at certain proportions. This study describes a highly efficient and stable method for the preparation of TTX from pufferfish liver, which can be useful for further analysis and/or research on TTX from various organisms.

## 4. Materials and Methods

### 4.1. Pufferfish

Wild pufferfish (*Tagifuju pseudommus*) was captured from the coast of East China Sea in Zhejiang Province. The identity of the fish was confirmed by fish experts from the Institute of Marine Fisheries, Zhejiang Province. Pufferfish liver was ground and mixed by a meat grinder; the resultant mixture was then stored at −20 °C until subsequent use.

### 4.2. Chemicals and Reagents

Chromatographic grade methanol and acetonitrile were purchased from Merck & Co., Inc. (Kennborough, NJ, USA) Chromatographic grade formic acid and ammonium acetate were purchased from Sigma-Aldrich Co. (Saint Louis, MO, USA) Analytical-grade solvents, including Na_2_HPO_4_, NaH_2_PO_4_, NaCl and NaOH were purchased from China National Pharmaceutical Group Corporation (Sinopharm, Shenzhen, China). All other reagents used in the study were of analytical grade.

### 4.3. UAE Method

Fifteen milliliters (15 mL) of methanol containing 1% acetic acid solution was mixed with 5.0 g of *T. pseudommus* liver. The mixture was then subjected to extraction by an ultrasonic (40 kHz) in a 60 °C water bath for 25 min. After a centrifugation at 6000 rpm, the supernatant (the extract) was transferred to a beaker. The pellet was further resuspended in 15 mL of methanol solution containing 1% acetic acid, and then centrifuged at 6000 rpm; and the resultant supernatant was transferred to a fresh beaker. The two extracts were combined and analyzed by UPLC-MS/MS.

### 4.4. UPLC-MS/MS Analysis

The analysis of TTX and its analogues in extracts of pufferfish liver were performed according to our previously reported method, with modification [27]. The experimental conditions were as follows. The analysis was carried out on a Waters ACQUITY UPLC I-Class system (Waters, Milford, MA, USA) coupled with a Xevo TQ-S triple-quadrupole mass spectrometer (Waters, Manchester, UK). UPLC separation was performed on an ACQUITY UPLC HILIC column (100 mm × 2.1 mm I.D., 1.7 mm particle size) at 40 °C, with an injection volume of 10 μL. Acetonitrile (mobile phase A) and 10 mmol·L^−1^ ammonium acetate in ultrapure water (pH 3.5, adjusted by formic acid) (mobile phase B) were used as the mobile phases; their flow rates were set at 0.3 mL·min^−1^ throughout the analysis. The gradient was set as follows (t is time and subscript numbers are the time in minutes):t_0_, B = 15; t_2.8_, B = 15; t_2.9_, B = 85; t_3.__4_, B = 85; t_3.5_, B = 15; t_6_, B = 15. The mass spectrometer was operated in positive ionization mode using an electrospray ion source; the capillary voltage was set at 3.0 kV. The source and the desolvation temperature were set at 110 °C and 350 °C, respectively. The cone and desolvation gas flow were 50 L·hr^−1^ and 600 L·hr^−1^, respectively. Data acquisition was conducted in multiple reactions monitoring mode (MRM), and related parameters for TTX and its analogues are shown in Table 3.

### 4.5. Experimental Design

#### 4.5.1. Single-Factor Selection Experiments

The liquid:material ratio, the extraction temperature, the extraction time, and the extraction times were selected for single factor optimization of TTX extraction process. The effect of liver:extraction solvent ratio (1:1, 1:2, 1:3, 1:4 and 1:5 (*w*/*v*)) on the TTX extraction yield was studied at an extraction temperature of 40 °C, an extraction time of 20 min, and an extraction time of 1. In addition, the effect of extraction time (20, 25, 30, 35, 40, 45, and 50 min) on the TTX extraction yield was investigated at a material:liquid ratio of 1:3, an extraction temperature of 40 °C, and with one extraction. Moreover, the effect of extraction temperature (20, 30, 40, 50, and 60 °C) on the TTX extraction yield was examined at a material:liquid ratio of 1:3, an extraction time of 20 min, and with one extraction. Finally, the effect of the number of extractions (1, 2, 3, 4, and 5) on the TTX extraction yield was investigated at a material:liquid ratio of 1:3, an extraction temperature of 40 °C, and an extraction time of 20 min.

#### 4.5.2. RSM Optimization Process

The single-factor optimization process indicated that the TTX extraction yield was unchanged although two cycles of extraction were performed. Therefore, the process with two cycles of extraction was further designed and optimized according to the central composite design principle of Box–Behnken [22], and statistical analysis software Design-Expert 8.0.6 (STAT-EASE Inc., Minneapolis, MN, USA) was used. The results from single-factor experiments (in which the material:liquid ratio, the extraction temperature, and the extraction time were set as the independent variables, and the TTX extraction yield was set as the response) were subjected to a three-factors-three-levels design, in which the codes for low, medium and high level of each independent variable were set to 1, 2, and 3, respectively. The levels and codes used in the experimental design are tabulated in Table 4.

### 4.6. Experimental Design

All data are averages of three repeated experiments, and the single factor experimental data were used to plot the trend curve using Origin 7.5 (OriginLab Corporation, Northampton, MA, U.S.A). Response surface experiments were performed using Design-expert 8.0.6 software for variance analysis, in which the optimal extraction process was optimized.

## Figures and Tables

**Figure 1 toxins-10-00529-f001:**
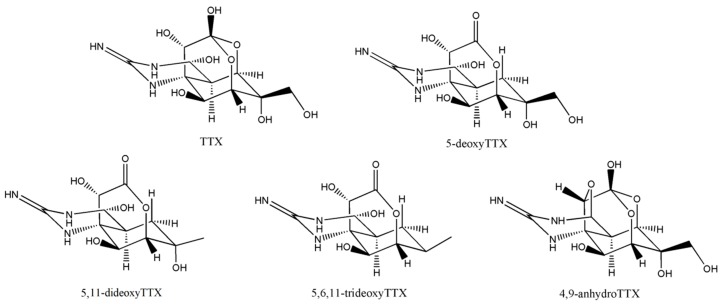
Chemical structure of dominant tetrodotoxin (TTX) analogues.

**Figure 2 toxins-10-00529-f002:**
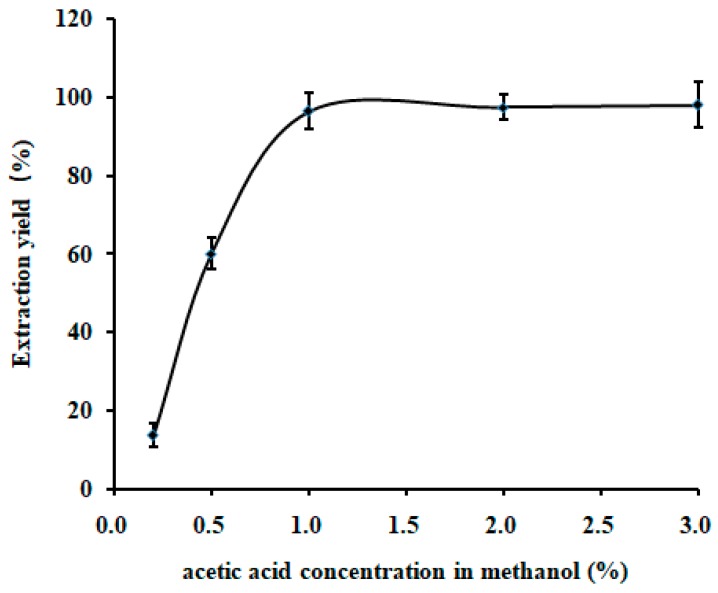
Effects of different acetic acid concentrations in methanol (0.2–3.0%) on the TTX extraction yield.

**Figure 3 toxins-10-00529-f003:**
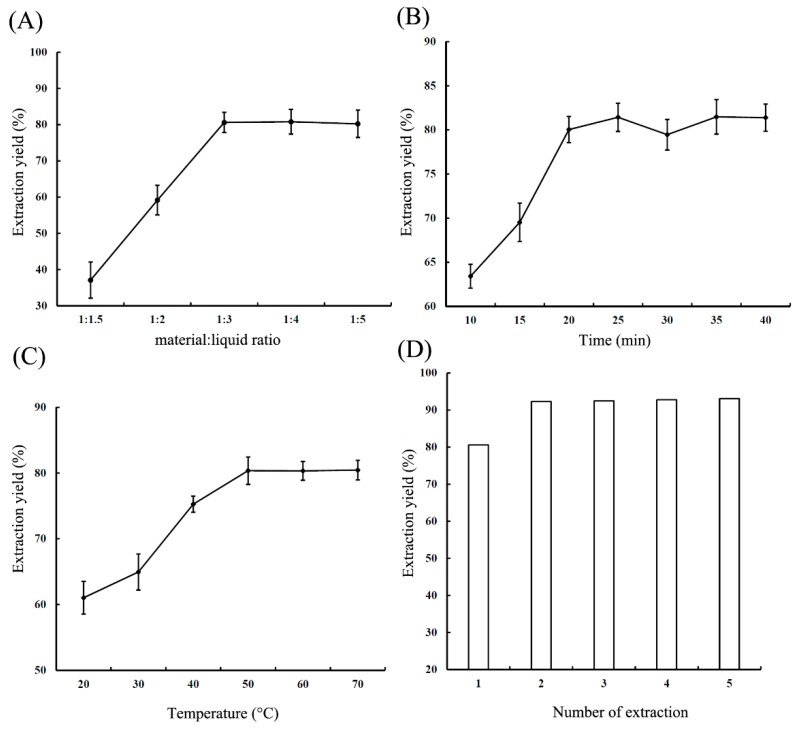
Effects of different extraction conditions on the TTX extraction yield: (**A**) material:liquid ratio; (**B**) extraction time; (**C**) extraction temperature; (**D**) number of extractions.

**Figure 4 toxins-10-00529-f004:**
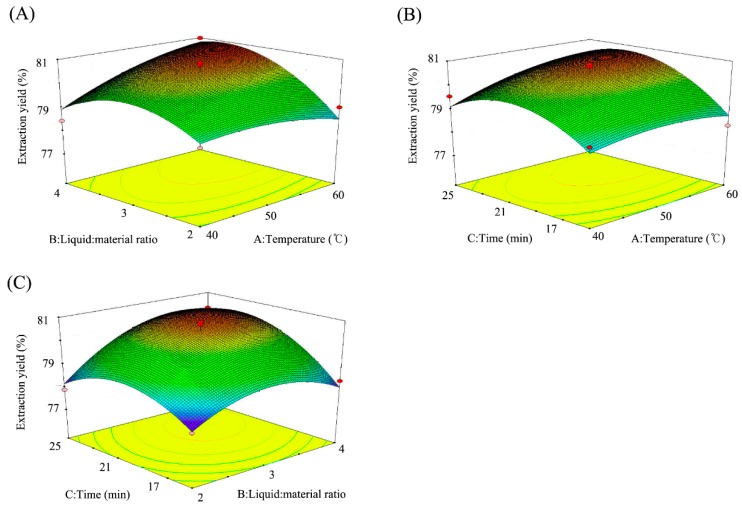
Response surface analysis of different factors: (**A**) liquid:material ratio and extraction temperature; (**B**) extraction time and temperature; (**C**) extraction time and material:liquid ratio.

**Figure 5 toxins-10-00529-f005:**
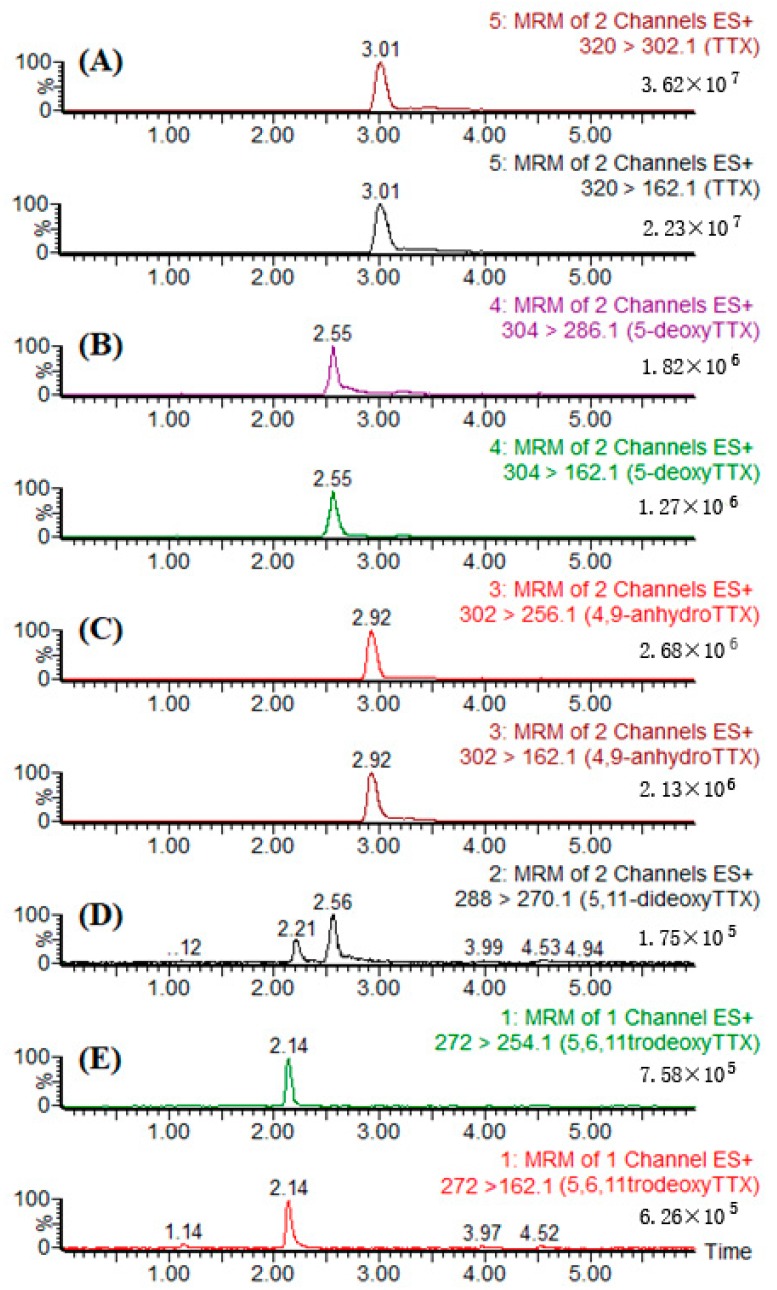
Monitored transitions of different TTX analogues extracted from the liver of *T. pseudommus*: (**A**) TTX; (**B**) 5-deoxy TTX; (**C**) 4,9-anhydro TTX; (**D**) 5,11-dideoxy TTX; (**E**) 5,6,11-trideoxy TTX. MRM, multiple reactions monitoring mode.

**Table 1 toxins-10-00529-t001:** The response surface methodology (RSM) experiments and results (TTX extraction yields).

Experiment Number	Temperature [°C]	Liquid:Material Ratio [mL·g^−1^]	Extraction Time [min]	TTX Extraction Yield [%]
1	0	1	1	80.3
2	1	0	1	80.25
3	0	0	0	80.01
4	−1	−1	0	78.78
5	0	−1	−1	77.94
6	−1	1	0	78.44
7	−1	0	1	79.55
8	0	1	−1	78.46
9	−1	0	−1	78.87
10	1	1	0	80.98
11	1	0	−1	78.31
12	1	−1	0	79.02
13	0	−1	1	77.89
14	0	0	0	80.02
15	0	0	0	80.87
16	0	0	0	80.76
17	0	0	0	80.88

**Table 2 toxins-10-00529-t002:** Initial regression analysis results.

Source	Sum of Squares	Degree of Freedom	Mean Square	*F* Value	*p* > *F*	Significant
Model	17.09	9	1.9	6.68	0.0102	Significant
A	1.07	1	1.07	3.75	0.0941	Insignificant
B	2.59	1	2.59	9.1	0.0195	Significant
C	2.43	1	2.43	8.55	0.0222	Significant
AB	1.32	1	1.32	4.65	0.068	Insignificant
AC	0.4	1	0.4	1.4	0.2761	Insignificant
BC	0.89	1	0.89	3.14	0.1197	Insignificant
A^2^	0.39	1	0.39	1.36	0.2822	Insignificant
B^2^	3.41	1	3.41	12	0.0105	Significant
C^2^	3.88	1	3.88	13.65	0.0077	Significant
Residual	1.99	7	0.28			
Lack of Fit	1.17	3	0.39	1.91	0.2698	Insignificant
Pure Error	0.82	4	0.2			
Total Deviation	19.09	16				

*F*, Fisher’s analysis of variance. *p*, probability value.

**Table 3 toxins-10-00529-t003:** Multiple reactions monitoring mode (MRM) parameters for TTX and its homologous.

Analyte	Precursor Ion (*m*/*z*)	Product Ion (*m*/*z*)	Cone Voltage (V)	Collision Energy (eV)
TTX	320	302.1 *	40	25
		162.1		35
5,6,11-trideoxyTTX	272	254.1 *	40	25
		162.1		
5,11-dideoxyTTX	288	270.1	40	25
4,9-anhydroTTX	302	256.1 *	40	25
		162.1		35
5-deoxyTTX	304	286.1 *	40	20
		162.1		25

* indicates quantitative ion.

**Table 4 toxins-10-00529-t004:** The factors and levels used in the response surface methodology (RSM) design for TTX extraction.

Factors	Levels		
	1	2	3
Extraction temperature (A) [°C]	40	50	60
Liquid:material ratio (B) [mL g^−1^]	2	3	4
Extraction time (C) [min]	15	20	25

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
