# Peer review of "Response Surface Methodology for the Optimization of Ultrasound-Assisted Extraction of Tetrodotoxin from the Liver of *Takifugu pseudommus"

_toxins, 2018, doi:10.3390/toxins10120529_

Reviewer 1 Report

I think this paper is not fit to "Toxins".

I think essential need required for toxic pufferfish liver is total toxicity. In order to determine total toxicity value of the liver, the official mouse assay is preferable. International Journal "Toxins" requires right person in the right place.

Author Response

I think this paper is not fit to "Toxins".

Response: The paper has been majorly revised, and hopefully its present form will be considered as appropriate to "Toxins" for publication.

Reviewer 2 Report

General comment

This study describes optimization of ultrasound-assisted extraction of tetrodotoxin from liver of Takifugu pseudommus. The optimization was done thoroughly, however, proportion of TTX analogues in the liver of Takifugu pseudommus is curious, arising a concern that TTX was decomposed to 5,6,11-trideoxyTTX during the extraction procedure. Otherwise the compounds could be wrongly identified. The proportion of TTX should be confirmed by traditional extraction procedure to verify validation of newly reported author’s extraction method, otherwise it could be difficult to be published in Toxins which has already been one of the leading Journal in this field. 

Detailed comments

Figure 1; Dominant TTX analogues should be shown in Figure 1.

p.1, l.6 the lack of efficient TTX extraction and preparation methods-;

There are already some excellent extraction methods for TTX. This sentence should be replaced with different description.

p.6, l.29; it was also discovered gastropods;

> it was also discovered from gastropods

p.2, l.68; The extraction process has been a bottleneck in the study of marine toxins;

I guess that the reference cited does not mean this. The extraction process could be a bottleneck of marine toxin analysis by instrumental methods including LC/MS.

p.2, l.70, Important factors affecting TTX extraction include material:liquid ratio;

One of the most important factors is solvent. It should be included. Why did the authors use methanol containing 1% acetic acid? It should be explained. I guess that the traditional solvent, acidic water containing acetic acid, was replaced to acidic methanol.

p.3, l.76; The experimental results showed that the extraction yield remained nearly constant-

The extraction yield is shown by percentages (%). These values should be calculated by comparing with extraction yield obtained by a traditional extraction method.

Figure 1; Dominant TTX analogues should be shown in Figure 1.

Homologues should be replaced to analogues.

Author Response

General comment

This study describes optimization of ultrasound-assisted extraction of tetrodotoxin from liver of Takifugupseudommus. The optimization was done thoroughly, however, proportion of TTX analogues in the liver of Takifugupseudommus is curious, arising a concern that TTX was decomposed to 5,6,11-trideoxyTTX during the extraction procedure. Otherwise the compounds could be wrongly identified. The proportion of TTX should be confirmed by traditional extraction procedure to verify validation of newly reported author’s extraction method, otherwise it could be difficult to be published in Toxins which has already been one of the leading Journal in this field.

Response: Related explain have been added in corresponding section. “Study showed TTX analogues distribution between either the tissues of the same organism or different specimens of the same species was unequal [28], While 5,6,11-trideoxyTTX was found to be the major analogue in all tissues of the pufferfish and gastropod [1, 29]. Considered the fact that the liver material was a combined sample consisted of different batches of pufferfish, our results were comparable to previous reports”, now at lines 167-171.

Detailed comments

Figure 1; Dominant TTX analogues should be shown in Figure 1.

Response: Revised as suggested. Structure for dominant TTX analogues is also provided in Fig.1.

p.1, l.6 the lack of efficient TTX extraction and preparation methods-;

There are already some excellent extraction methods for TTX. This sentence should be replaced with different description.

Response: Corresponding sentence has been deleted from the paper.

p.6, l.29; it was also discovered gastropods;

>it was also discovered from gastropods

Response: Revised as suggested.

p.2, l.68; The extraction process has been a bottleneck in the study of marine toxins;

I guess that the reference cited does not mean this. The extraction process could be a bottleneck of marine toxin analysis by instrumental methods including LC/MS.

Response: Corresponding sentence has been deleted from the paper.

p.2, l.70, Important factors affecting TTX extraction include material:liquid ratio;

One of the most important factors is solvent. It should be included. Why did the authors use methanol containing 1% acetic acid? It should be explained. I guess that the traditional solvent, acidic water containing acetic acid, was replaced to acidic methanol.

Response: As mentioned in our paper “It was traditionally and mainly used to extract TTX from pufferfish by using aqueous acetic acid solution[16,17]. However, since water is not easily evaporated or concentrated, this can subsequently bring a number of problems to the purification process of TTX, thereby can affect the extraction efficiency. It is necessary to improve and optimize the critical extraction process to promote the TTX production. According to our trail experiment, methanol containing 1% acetic acid served as an efficient substitute for acidified aqueous, yet systematic optimization has not been reported.”, now at lines 43-50.

p.3, l.76; The experimental results showed that the extraction yield remained nearly constant-

The extraction yield is shown by percentages (%). These values should be calculated by comparing with extraction yield obtained by a traditional extraction method.

Response: The present work aim to establish a high-efficiency TTX extraction method by optimizing the RSM, rather than compare the method performance. Related data may be provided in our further study.

Figure 1; Dominant TTX analogues should be shown in Figure 1.

Response: Revised as suggested. Structure for dominant TTX analogues is also provided in Fig.1.

Homologues should be replaced to analogues. 

Response: Revised as suggested. “homologues” was replaced by “analogues” in the whole paper.

Round  2

Reviewer 1 Report

Significantly revised

Author Response

Thank you for your attention, pleased see revised manuscript

Reviewer 2 Report

Unfortunately this study have not shown any data verifying validation of the method. At least extraction efficiency should be shown by comparing with the traditional extraction method.

Although the authors mention that methanol containing 1% acetic acid served as an efficient substitute for acidified aqueious according to their trial experiment, the data is not shown.

Unique toxin profile obtained by their method should be verified by comparing with toxin profile obtained by the traditional extraction method for the same sample specimen.

Author Response

(1) Unfortunately this study have not shown any data verifying validation of the method. At least extraction efficiency should be shown by comparing with the traditional extraction method

Response: revised as suggested, related information was provided in section 2.1, “The highly localized temperature and pressure provided by microwave assisted extraction can cause selective migration of target compounds from the material to the surroundings solution at a more rapid rate and higher extraction yield[24]. Compared with the traditional water-bath extraction method that utilize acidified aqueous solution[21], the new developed method have significantly shortened the extracting time from 70 min to 23min, as well as improved TTX yield by 22%”, now at lines162-167.

(2) Although the authors mention that methanol containing 1% acetic acid served as an efficient substitute for acidified aqueious according to their trial experiment, the data is not shown.

Response: revised as suggested, related information was provided in section 2.1, “For strong polar substance TTX, 0.1% acetic acid solution was extensively used to extract the analyte from biological samples[21]. However, high solubility of proteins and carbohydrates in water usually led to cloudy extracting solution, which was hard to be concentrated into a smaller volume, subsequently posing a challenge for further TTX purification. High percentage of organic reagent in the extraction system generally permits effective protein precipitation and carbohydrate removal. In this study, methanol was initially used as the substitution solution for water, and the effect of acetic acid concentration in methanol (%) on TTX extraction yield was evaluated. As shown in Fig.2, TTX recovery reached its maximum value of 96% when methanol contained 1% acetic acid. Therefore methanol containing 1% acetic acid was chosen as the ideal extraction solution in later experiment”, now at lines70-78.

To keep the text as concise as possible, the later sentences were reconstructed, “The extraction process has been a bottleneck in the study of marine toxins. The extraction of TTX can also be affected by a variety of complex factors, thereby needs to be investigated through systematic studies.Such extraction involves many conditions. Important factors affecting TTX extraction include material:liquid ratio, extraction time, extraction temperature and the number of extraction” was replaced by “The extraction of TTX can also be affected by a variety of important factors, including material:liquid ratio, extraction time, extraction temperature and the number of extraction. Thereby systematic study needs to be investigated through to find out most significant factor for TTX yield”, now at lines80-82.

(3) Unique toxin profile obtained by their method should be verified by comparing with toxin profile obtained by the traditional extraction method for the same sample specimen.
Response: revised. Major aim of our study  was to develop a high efficient extraction method for later separation and preparation of TTX and its analogues. Since the traditional method was tedious and time-consuming, extraction yield and MS confirmation was only carried out for the pilot scale of toxin extract during the initial phase of method comparison. Corresponding text have been provided as "Since the traditional method was tedious and time-consuming, MS confirmation was only carried out for the pilot scale of acidified aqueous extract during the initial phase of method comparison as mentioned above. TTX and other four analogues were also found in the aqueous extract, however the extraction yield for these five compounds experienced 22%-43% reduction", now at lines186-189.

 Besides, "Study has suggested that toxin in pufferfish is mainly due to TTX with a molecular weight of 320 [24]; thus this TTX becomes the main target of the present study. In recent years, some studies have reported the coexistence of various analogues of TTXin pufferfish[25,26]" was replaced by "Though recent study have proved abundant coexistence of various analogues of TTX[25,26]. Traditional method developed toxin extraction by targeting TTX as the main product[21], while lacking information on identification and confirmation for possible co-extraction of TTX analogues", now at lines.171-173.

Round  3

Reviewer 2 Report

The paper did not show any data of TTX contents and profiles of the sample obtained by the traditional extraction method by acidic water. It is very difficult to evaluate the author’s method. It is difficult to publish the paper.